# Estimation of N₂O Emissions from Agricultural Soils and Determination of Nitrogen Leakage

**Kristína Tonhauzer [1,2,\*], Peter Tonhauzer [2], Janka Szemesová [2] and Bernard Šiška [1]**

[1] Faculty of European Studies and Regional Development, The Slovak University of Agriculture in Nitra, Trieda Andreja Hlinku 609/2, 949 76 Nitra-Chrenová, Slovakia; bernard.siska@uniag.sk
[2] Slovak Hydrometeorological Institute, Jeséniova 17, 83315 Bratislava, Slovakia; peter.tonhauzer@shmu.sk (P.T.); janka.szemesova@shmu.sk (J.S.)
\* Correspondence: kristina.tonhauzer@shmu.sk; Tel.: +421-902-637-641

**Abstract:** Leaching of nitrogen from the soil is a natural but unfavorable effect that generates N₂O emissions. Exact quantification of nitrogen leakage is a challenging process. Intensive leakage occurs mainly when the soil is without vegetation and under specific climatic conditions. This paper aims to quantify the amount of nitrogen leakage from arable land and grassland, and to estimate N₂O emissions in 2017. Estimating the country-specific fraction of leached nitrogen ($Frac_{LEACH}$) is important for the emission balance from this source. Emissions are underestimated when the fraction is low; on the contrary, a high fraction causes overestimation. The internationally recognized fraction is 30%, according to the 2006 Intergovernmental Panel on Climate Control (IPCC) Guidelines. This factor represents the fraction of nitrogen losses compared to total nitrogen inputs and sources. In this study, we analyzed the effects of climatic conditions on agricultural soils in Slovakia to evaluate the area of nitrogen loss through leaching.

**Keywords:** nitrogen; leakage; N₂O; emissions; soils

## 1. Introduction

All forms of nitrogen emitted from the soil have an undeniable environmental impact on a regional and global scale. It is the subject not only of monitoring and scientific research but also of policy debates that result in national and international mitigation policies and measures. In particular, nitrogen emissions occur when the natural nitrogen cycle in the soil is disrupted. Excessive soil fertilization causes higher nitrogen loss, transfer into the environment, and altered water and air quality [1]. Irrigation and nitrogen fertilization play important roles in crop yield. An appropriate N application rate coupled with suitable irrigation schedules should be implemented to obtain high yields while reducing deep percolation and N leaching in intensively cropped farming practice [2].

Irrigation affects the water content in the soil profile and causes the movement of nitrates. Proper irrigation should not lead to nitrate leaching. Dissolved nitrates in the soil can increase depletion by plants and prevent flooding. Irrigation optimizes moisture conditions and the intensity of nitrification processes. Therefore, the potential for nitrate leaching from irrigated soil containing more nitrates is higher than in nonirrigated soil [1].

Leaching is an important part of the nitrogen cycle, with significant economic (crop yield) and environmental (eutrophication) impacts. Nitrogen leakage takes place due to its relatively large solubility in water and absorption due to the predominant negative charge of the solid phase of the soil [1].

Nitrates and nitrites cause increased contamination of surface and underground drinking water sources. The outcomes of water pollution by nitrates include human health risks and eutrophication of water, resulting in increased growth of cyanobacteria, algae, and higher plant forms [3].

The formation of gaseous nitrous oxide ($N_2O$) in the soil is primarily caused by microbial nitrification and denitrification and depends mainly on the concentration of N, soil water content, and temperature [4,5]. Soil denitrification takes place in the presence of a suitable reducing agent, usually organic carbon. For most conditions, when the oxidant amount is limited, nitrogen is reduced to the level of molecular $N_2$. If the availability of the oxidant outweighs the supply of the reducing agent, the reduction of nitrogen is incomplete, and the ratio of $N_2O/N_2$ increases. This means that the use of nitrogen fertilizers, which are needed for intensive farming, increases $N_2O$ emissions into the air [6].

Another source of emissions is the leaching and draining of nitrogen from the soil. It occurs when there is an excess of $NO_3^-$ and its fixation in the vegetation system does not occur. $NO_3^-$ is washed away by precipitation into surface streams. With the denitrification processes described above, $NO_3^-$ is transformed into $N_2O$ emissions, especially in underground sources contaminated with nitrogen oxides [7,8].

Indirect $N_2O$ emission from agricultural nitrogen leaching into water pathways contributes to the global atmospheric $N_2O$ budget. However, there remains high uncertainty regarding this source in the bottom-up $N_2O$ inventory [9]. The emission factor of indirect $N_2O$ associated with N leaching and runoff ($EF_5$; kg $N_2O$–N per kg of $NO_3$–N) incorporates three components: groundwater and surface drainage, rivers, and estuaries [9].

In terrestrial ecosystems, the additional nitrogen leads to enhanced amounts of nitrogen cycling between vegetation and soil, with the primary removal process by leaching as $NO_3$ to groundwater and denitrification as $N_2$ back to the atmosphere. Equation (1) shows the denitrification process [8]:

$$[NO_3]^- \rightarrow [NO_2]^- \rightarrow [NO] \rightarrow [N_2O] \rightarrow [N_2] \tag{1}$$

Plant and soil communities have evolved to sequester and recycle nitrogen, as it is an essential and often limiting nutrient [10].

Nitrogen leaching does not occur continuously and across the entire territory. The amount of precipitation must be higher than the evaporation, and the water content of the retention field capacity due to heavy rain or irrigation might temporally exceed field capacity [11]. The retention capacity of soil to retain water, or the ability to accumulate water, is one of the most important properties of soil and significantly influences the water cycle in nature and in the production capacity of the soil. This property depends on soil parameters, environmental characteristics, and soil depth. Soil characteristics are mainly grain size, mineralogical composition, quality and arrangement of soil horizons, quality of organic matter, structure, and content. Environmental features include surface topography, weather conditions, precipitation, and groundwater level [12].

Physical, chemical, and biological attributes of soil influence the leaching of nitrogen. The type and quality of soil determine the intensity of rainwater leakage with respect to irrigation water, and its capacity to retain water more or less prevents leakage. There is unquestionable evidence of increased nitrogen leaching from light, less humid, and shallow soils, and conversely, a barrier to leaching nitrogen in heavy, low permeable soils (clays) [13]. It is plausible that leaching is only possible when precipitation predominates over evapotranspiration or when groundwater temporarily enters the soil profile, especially in the spring [14]. According to the findings of Cardenas and his team, intense precipitation negate the effects of soil type on leaching, with most nitrate available for leaching being lost irrespective of soil type [13].

Fertilization is the most important anthropological factor of nitrogen leaching from the soil. Davies and Bradly [15] considered that increasing fertilizer doses leads to increased leaching intensity. The nitrogen dose, the fertilizer form, and the application time play important roles here. Nitrogen leaching can be significantly accelerated when fertilizers are applied in autumn and at increased doses [15].

## 2. Materials and Methods

According to Mosier et al. [16], the suggested $Frac_{LEACH}$ value is 30%. $Frac_{LEACH}$ represents the fraction of nitrogen losses in managed soils in regions where leaching occurs compared to total nitrogen inputs and sources [17]. This was recommended for calculation of $N_2O$ emissions through leaching in the 2006 Intergovernmental Panel on Climate Change (IPCC) Guidelines, which define that for areas with active irrigation and areas where total precipitation is higher than evaporation for a short time, the value of 30% of the proportion of nitrogen leached out of the utilized agricultural land ($Frac_{LEACH}$) should be used. For dryland regions, where precipitation and irrigation are lower than evapotranspiration throughout most of the year, leaching is unlikely to occur, and $Frac_{LEACH}$ is equal to zero [17].

Including irrigated and wet areas modifies the default nitrogen leached from arable land and grassland $Frac_{LEACH}$ to a national value according to the following equation:

$$Frac_{LEACH_N} = (Frac_{IRR} + Frac_{WET}) \times Frac_{LEACH} \tag{2}$$

where $Frac_{IRR}$ is the proportion of irrigated areas to total agricultural land area, $Frac_{WET}$ is the share of the wet area to the total area of arable land and grassland (%), and $Frac_{LEACH_N}$ is the national value of the proportion of leached nitrogen from cultivated soil (%).

### 2.1. Analysis of Irrigated Areas in Slovakia

The share of irrigated areas in Slovakia was derived from official statistics published by Hydromelioration, a state enterprise. The data were compared with Eurostat datasets. Identified data gaps and inconsistencies are shown in Table 1. The total area of utilized agricultural land was taken from the official statistics of the Statistical Office of the Slovak Republic. For the correct determination of the proportion of irrigated areas, distinguishing irrigation type was important. In the case of drip irrigation, water is gradually soaked into the soil and no nitrogen leaching occurs. Therefore, drip irrigation areas were excluded from the analysis [17]. From the statistics, it can be seen that the proportion of irrigated areas in Slovakia is decreasing due to the obsolescence of the irrigation network, with a decrease of 79.9% from 2002 to 2017. Statistical data concerning irrigated areas could not be fully verified because only Hydromelioration publishes this type of data in its annual reports; the Statistical Office did not publish such data, and Eurostat published only an incomplete proportion of irrigated areas (proportions are available for 2006, 2008, 2011, and 2014) based on its own methodology or estimation, with a lack of transparency.

For 2017, the total irrigated area in Slovakia was 54,421 hectares, representing only 3.6% of agricultural land [18]. According to Eurostat, the average in 28 European Union countries was 11.3%. Improving water efficiency and developing irrigation systems have been priorities of the Rural Development Program for 2014–2020. Farmers could apply for a nonrepayable financial contribution to restore their irrigation systems in 2017. An increase in the proportion of irrigated areas and the large year-on-year fluctuations in crop yield, which depend on climatic conditions, hence a lack of inadequate distribution of precipitation, can be expected in the future. The proportion of irrigated areas to total utilized agricultural areas is given in Table 1.

**Table 1.** Proportion of irrigated areas to total utilized agricultural areas.

| Year | Total Irrigated Areas (ha) | Utilized Agricultural Area (ha) | Share of Irrigated Areas to Total Areas of Agricultural Use (Frac$_{IRR}$) | Share of Irrigated Areas According to Eurostat |
|------|------|------|------|------|
| 2002 | 268,738 | 1,497,354 | 17.9% | |
| 2003 | 294,202 | 1,499,323 | 19.6% | |
| 2004 | 220,861 | 1,501,425 | 14.7% | |
| 2005 | 147,519 | 1,504,147 | 9.8% | |
| 2006 | 196,749 | 1,507,400 | 13.1% | 2.4% |
| 2007 | 226,548 | 1,507,698 | 15.0% | |
| 2008 | 225,436 | 1,507,278 | 15.0% | 2.0% |
| 2009 | 214,326 | 1,503,561 | 14.3% | |
| 2010 | 206,523 | 1,501,997 | 13.7% | |
| 2011 | 194,215 | 1,500,905 | 12.9% | 0.8% |
| 2012 | 187,574 | 1,499,568 | 12.5% | |
| 2013 | 168,277 | 1,498,986 | 11.2% | |
| 2014 | 154,698 | 1,498,119 | 10.3% | 1.3% |
| 2015 | 62,239 | 1,495,789 | 4.2% | |
| 2016 | 60,818 | 1,494,900 | 4.1% | |
| 2017 | 54,421 | 1,494,566 | 3.6% | |

## 2.2. Estimation of Wet Areas in Slovakia

The climatic parameters evapotranspiration and precipitation (Figure 1) were used to estimate wet areas in Slovakia. Detailed data were obtained from 41 regular meteorological stations (Figure 2) operated by the Slovak Hydrometeorological Institute (SHMI). Data were analyzed and aggregated to monthly and annual averages for this study.

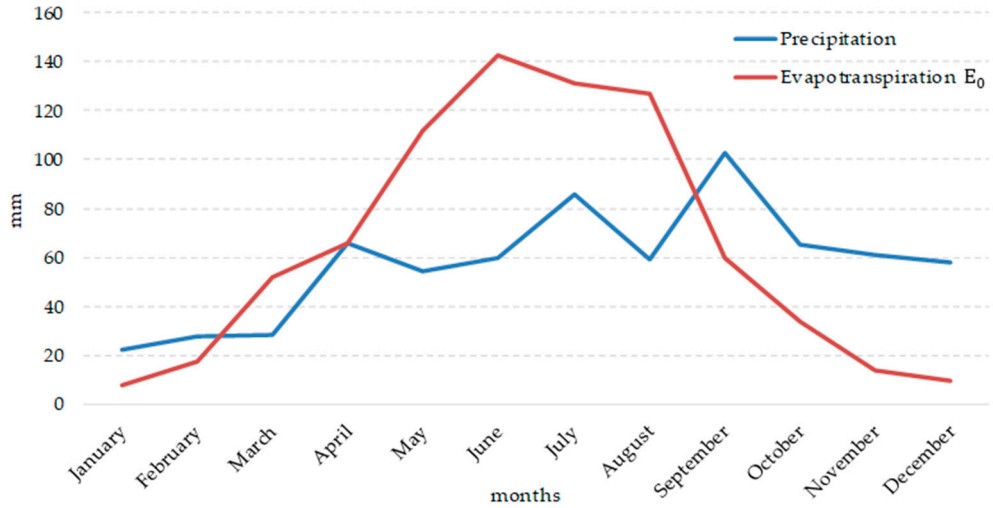

**Figure 1.** Monthly average of evapotranspiration and precipitation for the whole area in 2017.

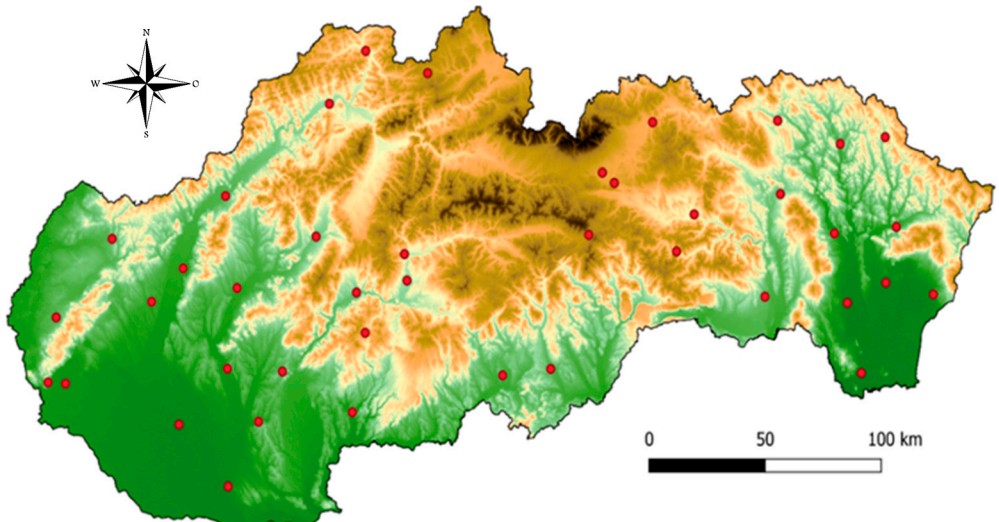

**Figure 2.** Network of meteorological stations in Slovakia.

Evaporation in agricultural areas occurs mainly through evapotranspiration ($ET_0$) and depends on meteorological conditions, soil characteristics, farming practices, and crop types. This means that evapotranspiration can vary within the country or over time and cannot be expressed by one single representative value. For the purposes of this study, we assumed the appearance of vegetation during the whole year, therefore we replaced evaporation with evapotranspiration [19].

Evapotranspiration was estimated for all 41 regular meteorological stations with the Penman–Monteith combined method [19]. The method combines the water–heat transport equation with Equation (3) as the energy conservation equation for the soil–plant–atmosphere system [19]. The reference surface is the area on which the reference crop grows, i.e., grass with specific properties (0.12 m height, surface resistance to water vapor transmission $r_s = 70$ s·m$^{-1}$, with albedo a = 0.23) [19]. The reference $ET_0$ concept was introduced to study water evaporation demand independent of crop type, crop development, and management practices. $ET_0$ values were calculated at different locations and seasons and are comparable because they refer to $ET_0$ from the same reference surface. The only factor influencing $ET_0$ is the climatic parameter. $ET_0$ expresses the evaporative power of the atmosphere at a particular location and time without taking into account crop characteristics and soil factors [19]:

$$ ET_0 = \frac{0.408\Delta(R_n - G) + \gamma \frac{900}{T+273} u_2(e_s - e_a)}{\Delta + \gamma(1 + 0.34u_2)} \tag{3} $$

where $ET_0$ is evapotranspiration (mm·day$^{-1}$), $R_n$ is net radiation at the crop surface (MJ·m$^{-2}$·day$^{-1}$), G is soil heat flux density (MJ·m$^{-2}$·day$^{-1}$), T is the daily air temperature at 2 m height (°C), $u_2$ is the wind speed at 2 m height (m·s$^{-1}$), $e_s$ is saturation vapor pressure (kPa), $e_a$ is actual vapor pressure (kPa), $e_s - e_a$ is saturation vapor pressure deficit (kPa), $\Delta$ is the slope of the vapor pressure curve (kPa·°C$^{-1}$), and $\gamma$ is the psychrometric constant (kPa·°C$^{-1}$).

The equation uses standard climatological data of solar radiation (sunshine), air temperature, humidity, and wind speed. The measurements were made at 2 m (or converted to that height) above an extensive surface of green grass completely shading the ground and with adequate water [20].

Aridity, a climatic indicator, is a climatological index used for regionalization of climate moisture conditions. It represents the relationship of the possible amount of water that can evaporate from the surface of weather-saturated soil and vegetation. The climatic index of aridity is calculated by the following equation [21]:

$$ \text{Aridity index} = \frac{P}{ET_0} \tag{4} $$

where Aridity index is defined by variables $ET_0$, the sum of monthly values of potential evapotranspiration for the deficient months in mm, and P, the sum of monthly values of total precipitation in mm.

The wet season must be identified for estimating wet areas. The rainy season is defined as the period when precipitation is higher than evapotranspiration. If the aridity index of the soil is greater than 1, the Equation (4) becomes [22]:

$$\frac{P}{ET_0} > 1 \qquad (5)$$

According to the definition of $Frac_{LEACH}$ in the 2006 IPCC Guidelines, the determination of rainy seasons is based on precipitation and pan evaporation ($E_{PAN}$) data. Rainy seasons are defined as periods when rainfall $> 0.5 \times$ pan evaporation, then $P/E_{PAN} > 0.5$, where P is monthly precipitation [17]. In the case of this study, we used evapotranspiration $\sum P/\sum ET_0 \geq 1$ [23]. The $P/ET_0$ share was analyzed for the 41 regular meteorological stations, and leaching was identified at 17 of them in 2017 (bold values in Table 2).

**Table 2.** Proportion of average annual precipitation in average reference evapotranspiration in 2017.

| Name of Station | Latitude | Longitude | ($P/ET_0$) |
|---|---|---|---|
| **Kuchyňa** | 48.40 | 17.12 | 0.6 |
| **Trenčín** | 48.88 | 18.05 | 0.7 |
| **Senica** | 48.69 | 17.40 | 0.7 |
| **Bratislava–Koliba** | 48.17 | 17.11 | 0.5 |
| **Bratislava–Airport** | 48.17 | 17.21 | 0.4 |
| **Jaslovské Bohunice** | 48.49 | 17.66 | 0.5 |
| **Žihárec** | 48.07 | 17.88 | 0.6 |
| **Piešťany** | 48.61 | 17.83 | 0.5 |
| **Žilina** | **49.23** | **18.61** | **1.2** |
| **Topoľčany** | 48.56 | 18.15 | 0.6 |
| **Podhájska** | 48.11 | 18.34 | 0.7 |
| **Nitra** | 48.28 | 18.14 | 0.5 |
| **Mochovce** | 48.29 | 18.46 | 0.7 |
| **Hurbanovo** | 47.87 | 18.19 | 0.6 |
| **Čadca** | **49.43** | **18.81** | **1.7** |
| **Prievidza** | 48.77 | 18.59 | 0.8 |
| **Oravská Lesná** | **49.37** | **19.18** | **2.5** |
| **Dudince** | 48.17 | 18.88 | 0.8 |
| **Banská Bystrica** | **48.73** | **19.12** | **1.2** |
| **Žiar nad Hronom** | 48.59 | 18.85 | 0.9 |
| **Banská Štiavnica** | **48.45** | **18.92** | **1.2** |
| **Sliač** | **48.64** | **19.14** | **1.0** |
| **Boľkovce** | 48.34 | 19.73 | 0.7 |
| **Poprad** | 49.07 | 20.25 | 0.9 |
| **Telgárt** | **48.85** | **20.19** | **1.5** |
| **Rimavská Sobota** | 48.37 | 20.01 | 0.8 |

**Table 2.** *Cont.*

| Name of Station | Latitude | Longitude | (P/ET$_0$) |
|---|---|---|---|
| Švedlár | 48.81 | 20.71 | 1.0 |
| Spišské Vlachy | 48.94 | 20.80 | 1.0 |
| Podolínec | 49.25 | 20.53 | 1.1 |
| Gánovce | 49.03 | 20.32 | 0.9 |
| Prešov | 49.03 | 21.30 | 0.8 |
| Bardejov | 49.28 | 21.27 | 1.3 |
| Čaklov | 48.90 | 21.63 | 0.9 |
| Košice | 48.67 | 21.24 | 0.6 |
| Tisinec | 49.21 | 21.65 | 1.2 |
| Medzilaborce | 49.25 | 21.91 | 1.7 |
| Milhostov | 48.66 | 21.72 | 0.8 |
| Somotor | 48.42 | 21.82 | 1.0 |
| Michalovce | 48.74 | 21.94 | 1.0 |
| Orechová | 48.71 | 22.22 | 1.2 |
| Kamenica nad Cirochou | 48.93 | 21.99 | 1.2 |

To cover the whole area of Slovakia, the meteorological data presented were interpolated and processed in a geographic information system (QGIS software) using the distance weighting interpolation function. Interpolation parameters were as follows: distance coefficient = 2, number of columns = 3000, and number of rows = 1500. The resulting map is shown in Figure 3. The red and orange parts of the map indicate places with no nitrogen leaching in 2017. The driest areas were in the lowlands (Danubian and Záhorská lowlands). In contrast, the yellow, green, and blue parts of the map show areas where nitrogen was leached (northern and central parts).

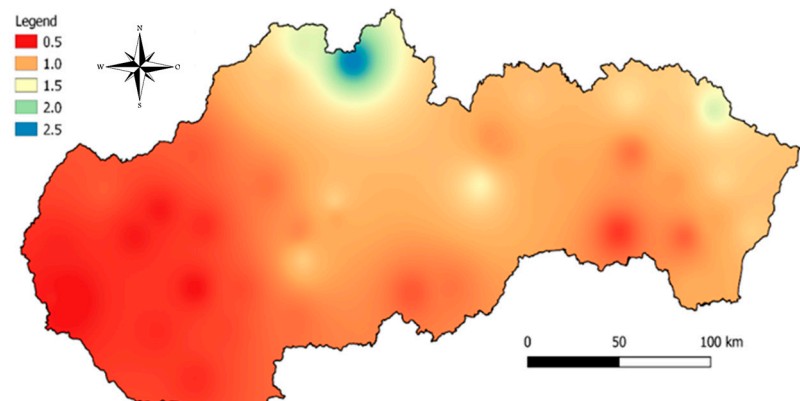

**Figure 3.** Ratio of average precipitation to average reference evapotranspiration (P/ET$_0$) in 2017.

The biggest deficit of moisture occurred mostly in the summer in the Danubian and Zahorská lowlands, where evaporation exceeded total precipitation by 60 mm on average. The deficit of precipitation is significant in lowlands in the yearly balance (in the Danubian lowland, it is 200 mm) according to moisture characteristics from the climate atlas of Slovakia for the period 1995–2010 [22].

In the raster image (Figure 3), areas with $\sum P / \sum ET_0 \geq 1$ were extracted by using the contour function and used to trim the underlying layers by available geoprocessing tools. A highly accurate database called the Land Parcel Identification System (LPIS) was used as an underlying layer. LPIS is

a part of the control mechanism under the Common Agricultural Policy [24]. It plays a significant role in verifying eligibility for area-based subsidies, monitoring farmers' cross-compliance with selected environmental rules, and managing the Rural Development Programmes [25].

### 2.3. Estimation of $N_2O$ Emissions from Leached Nitrogen

Agricultural soil, a significant source of nitrous oxide ($N_2O$) emissions in Slovakia, accounted for 72% of emissions in the country in 2017. $N_2O$ emissions from agricultural soil consist of direct emissions from the application of animal manure and fertilizer and indirect emissions from nitrogen leaching and runoff from ammonia and nitrogen oxides ($NH_3$ and $NO_x$) [26].

The accurate way to calculate $N_2O$ emissions in agriculture is based on nitrogen flow. Nitrogen is an essential element for livestock and crop growth. The main pathways of nitrogen from the soil are demonstrated in Figure 4. The agricultural sector has strongly altered nitrogen cycles. Nitrogen exceeding plant and animal needs may have a greater chance of being transferred to the atmosphere and aquatic ecosystems, thus the addition of N can result in increased nitrogen saturation in the environment [27].

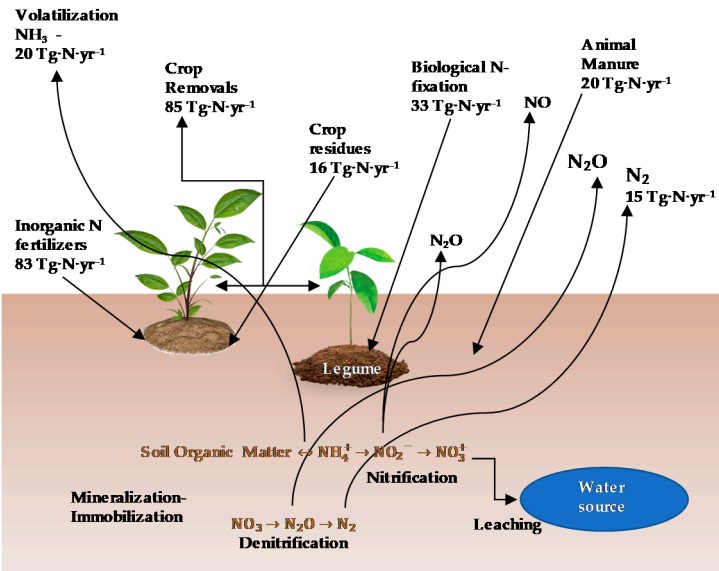

**Figure 4.** Simplified view of the nitrogen cycle in crop production. Estimated global N flows are quantified (inputs and losses, $Tg \cdot N \cdot yr^{-1}$) [28,29].

Part of the inorganic nitrogen in or on the soil, mainly in the form of $NO_3^-$, can bypass biological retention mechanisms in soil/vegetation systems by being transported in overland water flow or flow through soil macropores or pipe drains. Where more $NO_3^-$ is present in the soil than required by biological demand (e.g., under cattle urine patches), the excess leaches through the soil profile. Denitrification and nitrification transform some of the $NH_4^-$ and $NO_3^-$ to $N_2O$. This can take place in the groundwater where the N was applied (synthetic N fertilizer; organic N applied as fertilizer, e.g., applied animal manure, compost, sewage sludge; urine and dung N deposited on pasture; N in crop residues, including N-fixing crops and forage) [17].

The equation to calculate $N_2O$ emissions according to the 2006 IPCC methodology follows:

$$N_2O_{(L)} = \frac{(F_{SN} + F_{ON} + F_{PRP} + F_{CR}) \times Frac_{LEACH_N} \times EF_5}{1.57} \tag{6}$$

where $N_2O_{(L)}$ is the annual amount of gaseous $N_2O$ emissions in Gg per year during erosion and leaching; $F_{SN}$ is the annual amount of inorganic fertilizer applied to agricultural land in kg N per year; $F_{ON}$ is the amount of manure and slurry produced during livestock breeding, including compost

and sludge from sewage treatment plants, in kg N per year; $F_{PRP}$ is the amount of manure and slurry produced during pasture grazing; $F_{CR}$ is the amount of nitrogen in crop residues (above and below ground), including N-fixing crops from forage and pasture renewal, returned to soil annually in regions where leaching occurs, in kg N per year; $EF_5$ is the factor for $N_2O$ emissions from leaching; and 44/28 is the stoichiometric conversion factor for N to $N_2O$ [17].

$N_2O$ emissions from soil were calculated using Equation (6). A default emission factor, $EF_5 = 0.0075$ kg $N_2O$–N/kg N [17], and the national value for nitrogen loss in 2017 (7.86%) were used. Other inputs of nitrogen to agricultural land and grassland by activities in 2017 are shown in Table 3. The amount of nitrogen from synthetic fertilizers ($F_{SN}$) was taken from the Statistical Office of the Slovak Republic. The amount of organic nitrogen from compost was estimated from the total consumption of compost applied to agricultural land provided by the Central Control and Testing Institute in Agriculture [30]. Applied manure in soils is included in organic fertilizers ($F_{ON}$). Nitrogen left on agricultural land in the form of postharvest residues ($F_{CR}$) was estimated by using the 2006 IPCC methodology; nitrogen from grazing animals ($F_{PRP}$) was estimated in the same way. A detailed description of the calculation can be found in the 2019 National Inventory Report of the Slovak Republic 2019 [26].

**Table 3.** Nitrogen inputs to agricultural land in 2017, in tonnes.

| | |
|---|---|
| **Nitrogen from mineral fertilizers ($F_{SN}$)** | 122,541 |
| **Nitrogen from organic fertilizers ($F_{ON}$)** | 23,548 |
| **Nitrogen from postharvest residues ($F_{CR}$)** | 40,037 |
| **Nitrogen produced during grazing of farm animals ($F_{PRP}$)** | 8448 |
| **Total nitrogen applied to agricultural land** | 194,574 |

## 3. Results and Discussion

Leaching of nitrogen from agricultural land represents a considerable source of nitrogen emissions in Slovakia. A geoprocessing analysis of grassland and arable land data (Figure 5) revealed that in 2017 the total wet area was 524,875 ha, which was 22.6% of the total agricultural area ($Frac_{WET}$) and 10.9% of the total area of the Slovak Republic (Table 4). The total irrigated area ($Frac_{IRR}$) in Slovakia was 54,421 hectares, representing only 3.6% of total agricultural land. To calculate the specific national value of nitrogen loss from agricultural land due to leaching ($Frac_{LEACH_{NATIONAL}}$), we used Equation (7) with country-specific values of $Frac_{IRR} = 3.6\%$ (Table 1) and $Frac_{WET} = 22.6\%$. $Frac_{LEACH_{NATIONAL}}$ for the Slovak Republic was 7.86% in 2017 (Equation (9)).

$$Frac_{LEACH_N} = (Frac_{irr} + Frac_{wet}) \times Frac_{LEACH} \qquad (7)$$

$$Frac_{LEACH_N} = (3.6\% + 22.6\%) \times 30\% \qquad (8)$$

$$Frac_{LEACH_N} = 7.86\% \qquad (9)$$

**Table 4.** Share of wet areas in agricultural soils in 2017.

| Area | ha | Share of Total Agricultural Area | Share of Total Area of Slovakia |
|---|---|---|---|
| **Agricultural soils** | 1,972,260 | 100% | 47.4% |
| **Total wet areas** | 524,875 | 22.6% | 10.9% |

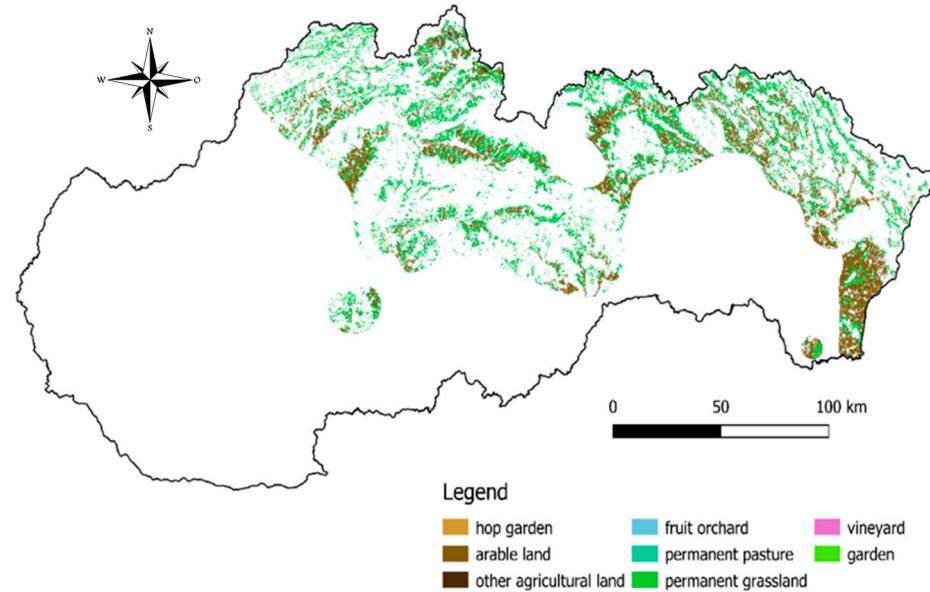

**Figure 5.** Grassland and arable land where $\sum P/\sum ET_0 \geq 1$.

Due to the heterogeneity of soil cover, nitrogen leaching is significant in some areas and minimal in others (Figure 5). In total, 7.8% of leached nitrogen in 2017 was from harvested fields and pastures, which represents almost 35.54 kg of nitrogen per hectare.

Using the national value of $Frac_{LEACH_N}$ instead of the default value from the IPCC 2006 Guidelines for the emissions estimation resulted in a decrease from 58.3 Gg to 18.7 Gg in 2017, or an almost 74% decrease (Table 5). $N_2O$ emissions from soil estimated using country-specific $Frac_{LEACH_N}$ reached 0.179 Gg.

**Table 5.** Total leached nitrogen, emission factor, and resulting $N_2O$ emissions from leaching in 2017.

| Results | Leached Nitrogen (kg) | Emissions of $N_2O$ (Gg) | Emission Factor (kg $N_2O$–N/kg N) |
|---|---|---|---|
| **Default Frac$_{LEACH}$** | 58,372,923 | 0.688 | 0.0075 |
| **National Frac$_{LEACH}$** | 18,653,779 | 0.179 | 0.0075 |
| **Emissions decrease compared to default Frac$_{LEACH}$ value** | −74% | | - |

This paper presents the development of the national methodology for estimating nitrogen leaching. The subsequent estimation of $N_2O$ emissions from agricultural soils and pastures for 2017 is shown. In a study conducted by Eder et al., 14% was shown as country-specific $Frac_{LEACH_N}$ for Slovakia [31], but without any background study.

According to the Bielek study, higher doses of nitrogen fertilizer on high-quality chernozem soil (Danubian lowland) have a negative impact on leaching. The difference between leaching of fertilized and nonfertilized soil is almost twofold: 45 kg N and 26 kg N per hectare, respectively. In this case, the applied dose was 100 kg N per hectare [32].

The current approach used in the national inventory of greenhouse gases (GHGs) calculates with the 2006 IPCC default value of the fraction of applied organic and inorganic N that is leached ($Frac_{LEACH}$ = 30%). The default value does not take into account the relationship between excess rainfall, soil type, and the timing of fertilizer and manure applications [33]. On the other hand, the Austrian value of 15.2% was experimentally measured by 22 lysimeters covering a wide range of soils, climatic conditions, and management practices [34]. The United Kingdom's average $Frac_{LEACH}$ value for both arable land and grassland is 18.0%, which is lower than the default IPCC value; the UK

also considers rainfall episodes and soil type in the calculation [13]. Ireland's average value of 10.0% was estimated from field measurements [34]. An analysis similar to ours was conducted in an Italian study by Costantini et al. [35], estimating areas with water surplus and irrigation. The water surplus was calculated by subtracting the evaporated water and the amount of water that can be retained in the soil (soil water holding capacity) from precipitation. This estimate was made at the level of $30 \times 30$ km mesh size, then a weighted average at the regional level was calculated. The weighted average value of $Frac_{LEACH}$ relative to the entire national agricultural area was 20.7% of the nitrogen applied to soils or deposited by grazing animals [35]. $Frac_{LEACH}$ parameters used in other European countries are shown in Table 6.

**Table 6.** Comparison of $Frac_{LEACHN}$ of selected EU countries published in 2017.

| EU Country | Austria * | Spain * | UK * | Italy * | Ireland * | Lithuania * | Slovakia |
|---|---|---|---|---|---|---|---|
| $Frac_{LEACHN}$ (%) | 15.2 | 8.3 | 18.0 | 20.7 | 10.0 | 23.0 | **7.9** |

* Source: [36].

As a result of this comparison, the $Frac_{LEACHN}$ presented in our study has the lowest value among European countries. This can be explained by the following observations: (a) the majority of agriculturally used land is located in the lowlands (Danubian and Záhorská lowlands), the driest areas of Slovakia (see Figure 3), where leaching did not occur in 2017; and (b) irrigation, which has a significant impact on $Frac_{LEACHN}$, is used minimally, due to obsolescent irrigation systems in Slovakia. These factors (climatic conditions and irrigation systems) are not comparable between countries, because a nation-specific fraction is used in the present paper.

Due to that fact that the Czech Republic and Poland, the closest neighboring countries, use default 2006 IPCC values for nitrogen leaching, they were not included in the comparison.

## 4. Conclusions

A default $Frac_{LEACH}$ value, representing the proportion of leached nitrogen, which was reported in the national 2020 GHG inventory submission, does not consider climatic and weather conditions in Slovakia and assumes that leaching exists on all agricultural soils in the country. This method significantly overestimates GHG emissions in this category. As this methodology does not consider national circumstances, an improved national leaching fraction value ($Frac_{LEACH_N}$) was developed in this study. The estimation of the $Frac_{LEACH_N}$ was processed by using a geographic information system (GIS). Base maps with the agricultural areas were taken from the LPIS. Isolines were processed from meteorological data on precipitation and evapotranspiration and applied to the base maps of the LPIS. The available data were interpolated to cover the area of Slovakia.

Improving the methodology for estimating emissions with the implementation of country-specific parameters is essential in reporting emissions from the agriculture sector. The utilization of $Frac_{LEACH_N}$ resulted in emissions decrease from 58.3 Gg to 18.7 Gg in 2017, a decrease of almost 74% (Table 5). This decrease is significant and represents a more accurate value of $N_2O$ emissions from nitrogen leaching.

Due to a lack of onsite measurements, the results presented in this study cannot be verified in practice. Case studies on field measurements are unavailable due to the lack of a reliable country network of lysimeters. Only old values before 1990 were available for a small part of Slovakia [32]. This can be improved in the future, and this study can accelerate the planning of possible research projects.

The improved emissions estimation will be implemented dynamically in the national emission inventory for the whole time series starting in 1990, which is the base year for emission estimations. Based on actual climatic parameters in the current year, the actual land database and the appropriate share of irrigated area, $Frac_{LEACH_N}$, will be calculated for each year and updated annually.

**Author Contributions:** Conceptualization, K.T. and P.T.; methodology, K.T.; software, P.T.; validation, K.T., P.T., J.S. and B.Š.; formal analysis, K.T.; investigation, K.T. and P.T.; resources, B.Š.; data curation, K.T.; writing—original

draft preparation, K.T. and P.T.; writing—review and editing, J.S.; visualization, K.T. and P.T.; supervision, B.Š.; project administration, K.T.; funding acquisition, B.Š. All authors have read and agreed to the published version of the manuscript.

**Funding:** This research was supported by the Scientific Grant Agency of the Ministry of Education, Science, Research and Sport of the Slovak Republic by project no. VEGA 1/0767/17.

**Conflicts of Interest:** The authors declare no conflict of interest.

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
