# Peer review of "Estimation of N2O Emissions from Agricultural Soils and Determination of Nitrogen Leakage"

_atmosphere, doi:10.3390/atmos11060552_

Round 1
Reviewer 1 Report
The authors have made extensive changes that have improved their paper. I recommend publication with the following very minor revisions.
Line 84: “Davies and S. Bradly consider” should be “Davies and Bradly consider”.
Line 283: Define GHG when it is first used.
Line 314: Lack of defining GHG makes Conclusion less clear.
Author Response
Response to Reviewer 1 Comments
Thank you very much for your recommendations.
Point 1: Point: Line 84: “Davies and S. Bradly consider” should be “Davies and Bradly consider”.
Response 1: Reference in the text of the manuscript was corrected
Point 2: Line 283: Define GHG when it is first used
Response 2: Information’s were included
Point 3: Line 314: Lack of defining GHG makes Conclusion less clear
Response 3: Information’s were included
Reviewer 2 Report
The paper was significantly revised. However, it continues to require some revision. The major problem of the manuscript is not good English. It is unfortunately not of publication quality and requires major improvement. I recommend providing very careful proof-read spell check to eliminate multiple stylistic and some grammatical errors. Native speaker manuscript spell-checking is preferable.
Specific comments.
Line 61 " The presented equation 1.1 presented ". Sentence needs revision
Line 67 close to zero ?
Line 67 " The retention capacity of the soil to retain water in ". Sentence needs revision
Line 72 weather conditions
Line 72 " relief, slope ". surface topography
Line 99 "FracLEACHNATIONAL " . Shorter name will be useful. e.g. FracLN
Lines 102-106 Too general sentence. Can be shifted to introduction
Line 128 areas ... areas...
Line 131 "humid area". I 'm not sure that it is correct term for areas with moist conditions
Line 133 " professional meteorological stations ". regular or mobile weather stations, I guess.
Line 134 operated by
Line 138 Sentence needs revision...
Evaporation and evapotranspiration are quite Synonyms.
Here you can say about contribution into evaporation of transpiration, soil evaporation, evaporation of intercepted water....
Line 142 replaced evaporation by what?
Line 144 " Aggregated values ". Monthly averaged for the whole area?
Line 145 " estimated by SHMI experts for ". It is not important info
Lines 145-146 " the Penman- Monteith combined method. ". The reference is required
Line 153 English editing is necessary
Line 155 In general form equation for ... can be written as ...
Line 161 climatological data
Line 162 the weather cannot be measured.
The measurements were made?
Lines 165-166 humidification ... humidification
Do you mean the climate moisture conditions?
Sentence is not clear
Line 165 As a climatic indicator
Line 167 " sufficiently humidified soil "... over moist.. water-saturated soils
Line 168 " humidification " ... moisture conditions
Line 169 " sum of ". for what period?
Line 170 soil moisture
Line 171 may be wet and dry seasons? not rainy
Line 173 The ratio of P and PET is actually the aridity index AI
Line 180 annual precipitation?
Lines 183-184 " number of columns 3000 and number of rows 1500 ". Sentence has to be reformulated
Line 183 Distance Weighting Interpolation method
Lines 185-187 more wet and dry areas were found or selected.... nitrogen analysis is done in the next step....
Line 191 humidity or moisture characteristics
Line 194 The ratio of
Line 202 Is it important for the method part?
Line 211 Look like the result....
Line 214-215 Look like results...... not method
Line 219 please reformulate the sentence
Line 220 " Inputs and outputs of ". The main pathways ...
Line 228-230 Can be easily shifted to Introduction.
Line 237 below the land? How?
Line 240 Reference?
Line 240 "44/28"... may be write it in decimal form?
Line 247 FracLEACHNATIONAL is was already explained in Eq. 1.2
Line 250 It is clear. You said already about it in line 240
Line 257 estimated using
Lines 283-284 English must be improved
Line 293 " the evaporated water by evapotranspiration ". Please reformulate
Line308 Czech Republic
Line 312 " FracLEACH " has to explained again
Lines 316-320 Such sentences are not for conclusion.
Line 317 determined in the study?
Author Response
Thank you very much for your recommendations. We implemented all recommendations and English editing by the native speaker was done.
Point 1: Line 61 " The presented equation 1.1 presented ". Sentence needs revision
Response 1: Sentence was revised
Point 2: Line 67 close to zero ?
Response 2: Sentence was refined
Point 3:Line 67 " The retention capacity of the soil to retain water in ". Sentence needs revision
Response 3: Sentence was revised
Point 4:Line 72 weather conditions
Response 4: Sentence was refined
Point 5: Line 72 " relief, slope ". surface topography
Response 5 : Sentence was refined
Point 6: Line 99 "FracLEACHNATIONAL " . Shorter name will be useful. e.g. FracLN
Response 6: Index was corrected
Point 7: Lines 102-106 Too general sentence. Can be shifted to introduction
Response 7: Paragraph was shifted to the introduction
Point 8: Line 128 areas ... areas...
Response 8: the redundant word was corrected
Point 9: Line 131 "humid area". I 'm not sure that it is correct term for areas with moist conditions
Response 9: humid area was exchange to wet
Point 10:Line 133 " professional meteorological stations ". regular or mobile weather stations, I guess.
Response 10: Proposed change implemented
Point 11: Line 134 operated by
Response 11: Proposed change implemented
Point 12: Line 138 Sentence needs revision...Evaporation and evapotranspiration are quite Synonyms. Here you can say about contribution into evaporation of transpiration, soil evaporation, evaporation of intercepted water....
Response 12: Proposed change implemented
Point 13:Line 142 replaced evaporation by what?
Response 13: Sentence was deleted, due to insignificance.
Point 14:Line 144 " Aggregated values ". Monthly averaged for the whole area?
Response 14: Explanation included
Point 15:Line 145 " estimated by SHMI experts for ". It is not important info
Response 15: Information was deleted
Point 16:Lines 145-146 " the Penman- Monteith combined method. ". The reference is required
Response 16: Reference was included
Point 17:Line 153 English editing is necessary
Response 17: Editing was done
Point 18:Line 155 In general form equation for ... can be written as ...
Response 18: Editing was done
Point 19:Line 161 climatological data
Response 19: Included
Point 20:Line 162 the weather cannot be measured. The measurements were made?
Response 20: Unclear wording corrected
Point 21:Lines 165-166 humidification ... humidification
Response 21: Corrected
Point 22: Do you mean the climate moisture conditions?
Response 22: Yes
Point 23: Sentence is not clear
Response 23: Unclear wording corrected
Point 24:Line 165 As a climatic indicator
Response 24: Corrected
Point 25:Line 167 " sufficiently humidified soil "... over moist.. water-saturated soils
Response 25: Corrected
Point 26:Line 168 " humidification " ... moisture conditions
Response 26: Corrected
Point 27:Line 169 " sum of ". for what period?
Response 27: Implemented
Point 28:Line 170 soil moisture
Response 28: Implemented
Point 29:Line 171 may be wet and dry seasons? not rainy
Response 29: Corrected
Point 30: Line 173 The ratio of P and PET is actually the aridity index AI
Response 30: Corrected
Point 31: Line 180 annual precipitation?
Response 31: Yes
Point 32:Lines 183-184 " number of columns 3000 and number of rows 1500 ". The sentence has to be reformulated
Response 32: Editing was done
Point 33: Line 183 Distance Weighting Interpolation method
Response 33: Corrected
Point 34: Lines 185-187 more wet and dry areas were found or selected.... nitrogen analysis is done in the next step....
Response 34: Corrected
Point 35:Line 191 humidity or moisture characteristics
Response 35: Corrected
Point 36: Line 194 The ratio of
Response 36: Corrected
Point 37: Line 211 Look like the result....
Response 37: Yes Shifted into result part
Point 38:Line 214-215 Look like results...... not method
Response 38: Shifted into result part
Point 39:Line 219 please reformulate the sentence
Response 39: Editing was done
Point 40:Line 220 " Inputs and outputs of ". The main pathways ...
Response 40: Editing was done
Point 41:Line 228-230 Can be easily shifted to Introduction.
Response 41: Lines were shifted to Introduction
Point 42:Line 237 below the land? How?
Response 42: Unclear wording corrected
Point 43:Line 240 Reference?
Response 43: Reference was included
Point 44:Line 240 "44/28"... may be write it in decimal form?
Response 44: Corrected
Point 45:Line 247 FracLEACHNATIONAL is was already explained in Eq. 1.2
Response 45: Explanation was deleted
Point 46:Line 250 It is clear. You said already about it in line 240
Response 46: Explanation was deleted
Point 47:Lines 283-284 English must be improved
Response 47: English editing was done
Point 48:Line 293 " the evaporated water by evapotranspiration ". Please reformulate
Response 48: Corrected
Point 49: Line308 Czech Republic
Response 49: Corrected
Point 50:Line 312 " FracLEACH " has to explained again
Response 50: Implemented
Point 51:Lines 316-320 Such sentences are not for conclusion.
Response 51: Corrected
Round 2
Reviewer 2 Report
The manuscript was significantly revised and it can be accepted for publication in J. Atmosphere. There are a few very small remarks to the manuscript that have to be answered before final acceptance.
- Line 95. FracLEACH is actually determined in abstract. However, I believe, it is necessary to explain this acronym once more within the main manuscript body.
- Line 167, Eq. 1.4. I'm not sure that the equation is correctly written. Aridity index is usually calculated as P/ET or P/PET
- Line 167. "humidification of soil moisture" is not the best term. May be to use the surface moisture?
Author Response
Thank you very much for your quick reply, very appreciated that.
Point 1: Line 95. FracLEACH is actually determined in abstract. However, I believe, it is necessary to explain this acronym once more within the main manuscript body.
Response 1: The reviewer has the truth. We recognize the lack of transparency concerning on acronym FracLEACH in the manuscript body. It was implemented directly in 95 line.
Point 2: Line 167, Eq. 1.4. I'm not sure that the equation is correctly written. Aridity index is usually calculated as P/ET or P/PET
Response 2: We changed the reference of the equation, now is equation correctly written.
Point 3: Line 167. "humidification of soil moisture" is not the best term. May be to use the surface moisture?
Response 3: The equation of arid index was corrected according to the web page: http://glossary.ametsoc.org/wiki/Aridity_index , description of K was deleted

This manuscript is a resubmission of an earlier submission. The following is a list of the peer review reports and author responses from that submission.
Round 1
Reviewer 1 Report
Peer review report on “Estimation of N2O Emissions from the Agricultural Soils and Determination of Nitrogen Leakages”
Initial Submission
This reviewer recommends a thorough revision or re-submission of this manuscript to be considered for publication in “Atmosphere”
Overview and general recommendation:
Nitrogen leaching is a natural phenomenon and its percolation into groundwater is a serious environmental concern. Nitrogen leaching potential and the chances of it entering the groundwater depend on precipitation, amount of water soluble Nitrogen in the soil, vegetation cover, water holding capacity of soil, the underlying zone structure (soil layer and/or bedrock), and the depth of water table. For this reason, in order to establish effective environmental and public health policies, extensive and detailed long-term case studies involving observation and numerical simulations are extremely important.
This reviewer has multiple major and minor concerns about publishing this manuscript in its current form
Major comments:
The authors attempt to derive a country specific N2O emission share from agricultural soils, as the internationally recognized share of 30% may not represent country specific values. However, these country-specific value has to be based on extensive measurements and these the country-specific value should be evaluated for its appropriateness. Introduction section must include relevant previous studies (and cite them) as a background to the current study. It should also major findings from previous studies.
To this reviewer, the current study should be expanded to be considered to publish as a journal article. Also, results from the study should be evaluated against measurements to ensure the correctness of the study.There are unnecessary discussions of basic science but very limited discussions are provided on the methodology and approaches specific to this study.
If you remove redundant information from line 205 -216 you will see there is hardly couple of sentences in the section “Methodology”. A thorough re-writing is needed for this article. Similarly the “results” section too lack details and additional studies and comparisons and discussions are needed. This reviewer does not follow the relevance of the section “Data verification” here. A section for result evaluation/validation has to be included. Conclusions merely lists already given information except that how this current estimation is going to be implemented into the national emission inventory. A thorough rewriting and updates are required in the “Methodology”, “Results and Discussions”, “Validation” and “Conclusions” sections. Abstract is a poorly written and suggest rewriting. Authors please move the data specification details in the abstract to “materials and methods” section. Authors have not referenced relevant previous observation or numerical modeling studies on air pollution during heatwave episodes under similar meteorological and geographical conditions. Lack of thorough literature review is clearly seen in the references section. There are nine articles referenced in this manuscript of which four of them are inventory reports and only five three of them are scientific studies.
Author Response
Point 1: The authors attempt to derive a country-specific N2O emission share from agricultural soils, as the internationally recognized share of 30% may not represent country-specific values. However, these country-specific value has to be based on extensive measurements and these the country-specific value should be evaluated for its appropriateness.
Response 1: The case study on field measurements is unavailable due to the lack of a reliable country network of lysimeters only old country studies before 1990 was available.
Point 2: To this reviewer, the current study should be expanded to be considered to publish as a journal article. Also, results from the study should be evaluated against measurements to ensure the correctness of the study. There are unnecessary discussions of basic science but very limited discussions are provided on the methodology and approach specific to this study.
If you remove redundant information from line 205 -216 you will see there is hardly a couple of sentences in the section “Methodology”. A thorough re-writing is needed for this article. Similarly, the “results” section to lack details and additional studies and comparisons and discussions are needed. This reviewer does not follow the relevance of the section “Data verification” here. A section for result evaluation/validation has to be included. Conclusions merely lists already given information except that how this current estimation is going to be implemented into the national emission inventory. A thorough rewriting and updates are required in the “Methodology”, “Results and Discussions”, “Validation” and “Conclusions” sections.
Response 2: we are aware that the lack of relevant previous and current studies in the manuscript was presented. During the second submission, mentioned deficiency would be corrected.
Point 3: Abstract is a poorly written and suggest rewriting.
Response 3: We agree with the statement. The abstract will be improved.
Point 4: Lack of thorough literature review is clearly seen in the references section. There are nine articles referenced in this manuscript of which four of them are inventory reports and only five three of them are scientific studies.
Response 4: Number of references will be extended
Point 5: Authors have not referenced relevant previous observation or numerical modelling studies on air pollution during heatwave episodes under similar meteorological and geographical conditions.
Response 5: Modelling studies on air pollution during heatwave episodes under similar meteorological and geographical conditions will be implemented.

Reviewer 2 Report
The authors present a study that quantifies the loss of nitrogen from arable lands and they estimated N2O emissions that result from farming, irrigation and rainfall in Slovakia.
It appears that 2017 is the target year for their final result but the abstract states “20[7” with a bracket appearing in place of “1” (page 1, line 17). This error points to the major problems with this paper. It is very poorly organized by content; it needs significant editing to make the language readable; and it’s not clear that the results are very important from their discussion and conclusions. I give a couple of examples below.
Poor Organization Examples
Equations 1.1 and 1.10 are identical. There is no need to repeat the equation twice.
The methodology section is not uniformly presented. First there is (Page 2, line 85) 2. “Materials and Methods” and then there is (Page 7, line 201 2.3.2. Methodology). Please correct.
It is not clear that the authors have conducted a valid study. For example (Page 3, line 121 – 122) the authors state that “Irrigation information was not available for the period 1990-2002, so time series was completed using linear extrapolation.” what data did the authors use for their extrapolation?
In conclusion, the authors may have some interesting results, but the results are obscured by a poorly organized paper and presentation. The paper might be publishable in the future provided authors make very major revisions.
I would be happy to review it again if it is revised and resubmitted.
Author Response
Response to Reviewer 2 Comments
Point 1: The authors present a study that quantifies the loss of nitrogen from arable lands and they estimated N2O emissions that result from farming, irrigation and rainfall in Slovakia.
Response 1: Thank you very much for your observation.
Point 2: Poor Organization Examples
Equations 1.1 and 1.10 are identical. There is no need to repeat the equation twice.
Response 2: Repetitive and redundant equation will be removed.
Point 3: The methodology section is not uniformly presented. First, there is (Page 2, line 85) 2. “Materials and Methods” and then there is (Page 7, line 201 2.3.2. Methodology). Please correct.
Response 3: During the second submission, mentioned deficiency would be corrected.
Point 4: It is not clear that the authors have conducted a valid study. For example (Page 3, line 121 – 122) the authors state that “Irrigation information was not available for the period 1990-2002, so time series was completed using linear extrapolation.” what data did the authors use for their extrapolation?
Response 4: We used values of irrigation only for the 2017 year. Therefore our opinion is that time-series for irrigation is redundant for our article. During the second submission, redundant information will be removed.
Point 6: In conclusion, the authors may have some interesting results, but the results are obscured by a poorly organized paper and presentation. The paper might be publishable in the future provided authors make very major revisions.
Response 6: We agree with this statement
